# Visible-Range Multiple-Channel Metal-Shell Rod-Shaped Narrowband Plasmonic Metamaterial Absorber for Refractive Index and Temperature Sensing

**DOI:** 10.3390/mi14020340

**Published:** 2023-01-28

**Authors:** Chung-Ting Chou Chao, Muhammad Raziq Rahimi Kooh, Chee Ming Lim, Roshan Thotagamuge, Abdul Hanif Mahadi, Yuan-Fong Chou Chau

**Affiliations:** 1Department of Optoelectronics and Materials Technology, National Taiwan Ocean University, Keelung 20224, Taiwan; 2Centre for Advanced Material and Energy Sciences, Universiti Brunei Darussalam, Tungku Link, Gadong BE1410, Brunei; 3Department of Nano Science Technology, Faculty of Technology, Wayamba University of Sri Lanka, Kuliyapitiya 60200, Sri Lanka

**Keywords:** multiple resonance modes, plasmonic metamaterial absorber, visible range, temperature sensitivity

## Abstract

Multiple resonance modes in an optical absorber are necessary for nanophotonic devices and encounter a challenge in the visible range. This article designs a multiple-channel plasmonic metamaterial absorber (PMA) that comprises a hexagonal arrangement of metal-shell nanorods in a unit cell over a continuous thin metal layer, operating in the visible range of the sensitive refractive index (RI) and temperature applications. Finite element method simulations are utilized to investigate the physical natures, such as the absorptance spectrum, magnetic flux and surface charge densities, electric field intensity, and electromagnetic power loss density. The advantage of the proposed PMA is that it can tune either three or five absorptance channels with a narrowband in the visible range. The recorded sensitivity and figure of merit (*S*, *FOM*) for modes 1–5 can be obtained (600.00 nm/RIU, 120.00), (600.00 nm/RIU, 120.00 RIU^−1^), (600.00 nm/RIU, 120.00 RIU^−1^), (400.00 nm/RIU, 50.00 RIU^−1^), and (350.00 nm/RIU, 25.00 RIU^−1^), respectively. Additionally, the temperature sensitivity can simultaneously reach 0.22 nm/°C for modes 1–3. The designed PMA can be suitable for RI and temperature sensing in the visible range.

## 1. Introduction

Metamaterials are artificial materials composed of nanometer-sized hybrid metal-dielectric nanostructures which can attain unique electromagnetic (EM) characteristics that cannot be found naturally [1,2,3]. Plasmonic metamaterials can be used to trap the EM wave in the nanometer regime and beyond the diffraction limit of traditional optics [4,5,6]. The surface of plasmonic metamaterials, i.e., metasurfaces, are two-dimensional (2D) arrays formed by an artificially designed periodicity array subwavelength metamaterial [7,8] based on the properties of localized surface plasmon resonance (LSPR) and Fabry–Pérot cavity resonance [9,10,11]. 

A plasmonic metamaterial absorber (PMA) is a potential optical device that has drawn attention due to its unique absorption features [12,13]. A PMA can be applied in nanophotonic fields, such as solar cells, photodetectors, sensors, thermal emitters, photoswitches, modulators, infrared spectroscopy, cloaking, and LEDs [14,15,16,17,18]. A microwave frequency band of a PMA was first proposed by Landy et al. in 2008 [19]. The absorption enhancement of an incident EM wave in a PMA is attributed to different patterns of mode resonances, e.g., LSPR, surface lattice resonance (SLR) [20,21,22], surface plasmon polariton resonance (SPPR) [23,24,25], gap plasmon resonance (GPR), and cavity plasmon resonance (CPR) [26,27,28]. After then, a variety of works were proposed to concern the absorptance response of a PMA working at the frequency range from a microwave [29] through terahertz (THz) [30], infrared [31,32], and into the visible region [33,34,35]. Chen et al. presented a THz PMA composed of a fan-like metasurface that can excite three absorptance peaks in the THz band [36]. M. Askari designed a gold slab array of PMA structures and obtained a maximum sensitivity and FOM of 1400 nm/RIU and 28.57, respectively, in the infrared region [37]. Wu et al. developed a PMA in the visible region by combining metal rods with suitable metal/dielectric layer nanocomposites [33]. Although a high absorptance can be achieved, only one absorptance peak exists in the visible range. 

PMA is a pivotal branch of optical elements in the optical spectrum due to its outstanding characteristics [38,39]. PMAs can commonly be categorized into broadband and narrowband [40,41]. PMAs concern their absorptance mode and full width at half maximum (FWHM) spectrum [42]. The absorption in a PMA is usually one or more narrowbands because of the nature of the resonances. For biosensing applications, a narrower FWHM (or band) can enhance the performance of sensitivity (*S*) and the figure of merit (*FOM*). It contains many exotic EM properties, which have many potential applications in nanophotonics. To improve sensor performance, many structures have been designed on metallic SPP-based subwavelength systems, including grooves [43,44], nanohole arrays [45], nanorods [46,47], nanorings [29,48], nanodisks [21,30], and so on. Callewaert et al. designed a PMA structure based on a dielectric nanoring array on the Ag layer with absorption intensities as large as 95% [34]. P. Mandal demonstrated a PMA made of a thin metal layer with an array of cylindrical grooves with 100% absorption [49]. Tittl et al. fabricated a palladium-based PMA and demonstrated a reflectance of <5% and zero transmittance at 650 nm [50]. However, the abovementioned PMAs have only one plasmonic mode in the visible range, thus limiting their application. 

In nanophotonics, multiple channels of resonance modes in a PMA are required and face a challenge in the visible range [51]. The multiple resonance metasurface sensors have the merit of more channels of working wavelengths compared to the single one in the visible wavelength [52,53,54,55]. Arising from the abovementioned essential topics, in this work, we propose and design a periodic array of PMA with a hexagonal arrangement of Ag-shell nanorods in a unit cell for refractive index (RI) and temperature sensing in the visible range. The Ag-shell nanorods have the function of trapping EM fields by fitting the impedance of free space. The Ag thin layer behind the impedance-matched layer can absorb the incident EM wave. Light impedance depends on the ratio between relative permittivity and permeability, and it is similar in concept to matching the load impedance in a circuit yielding maximum power transfer [56,57]. We study and compare two cases of PMAs, i.e., the solid Ag nanorods case (named case 1) and the Ag-shell nanorods case (called case 2). The case 2 PMA structure combines LSPR, CPR, and GPR effects and creates a vital SPP source in the plasmonic system. We utilize the finite element method (FEM) to simulate the absorptance spectrum, electric field (E-field) and surface charge density distributions, magnetic flux density, structural parameters, and electromagnetic power loss density profiles. The physical origin of multiple-channel absorptance peaks is associated with Fabry–Pérot resonance and the plasmonic effect of the Ag-shell nanorods in the PMA. The PMA is also applicable in RI and temperature sensing and implements three- or five-band channels in the visible range. Our designed PMA is promising for RI and temperature sensing applications in the visible range.

## 2. Simulation Method and Fundamental 

Figure 1a,b displays the unit cells of the investigated PMAs (case 1 and 2) with periodic arrays and a hexagonal arrangement. The proposed structures are composed of solid Ag nanorods (case 1) and Ag-shell nanorods with air core (case 2), deposited on an Ag thin layer with a substrate of silica (SiO_2_). The semi-spherical heads are placed on the top of the solid Ag/Ag-shell nanorods. As shown in Figure 1, the geometrical parameters are the pitch (*P_x_* and *P_y_*), the height of solid Ag/Ag-shell nanorods (*h*), the thickness of Ag (*t_Ag*), the outer and inner radius of Ag shell nanorods (*r_1_* and *r_2_*, *r_1_* − *r_2_* = *t_Ag*), the radius of the semi-spherical head (*r_1_*), the thickness Ag layer (*t_Ag*) at the bottom, the angle of incident light (*θ,* the angle between the z-axis in the *x-z* plane), and the polarization angle of incident light (*φ,* the angle between x-axis in *x-y* plane). Note that the semi-spherical head is solid and placed on the top of the solid Ag/Ag-shell nanorods. We compare the absorptance spectra of a hemisphere on top of the nanorods and a flat top of the nanorods and found that the former can achieve a higher absorptance peak and a narrower FWHM. Therefore, we place a hemisphere on top of the nanorods.

Simulations use FEM-based COMSOL Multiphysics with periodic boundary conditions at the side surfaces of the unit cell to imitate the periodic array of PMA. Perfectly matching layers (PMLs) are placed at the top and bottom surfaces of the unit cell and scattering boundary conditions (SBCs) are set on the PMLs’ surfaces to prevent the reflection of light at boundaries [58,59]. Under normal incidence, the x-polarization or y-polarization incident EM wave are coupled with the fundamental SPP mode impinges from the top surface of the unit cell (port 1) to the bottom surface of the unit cell (port 2). We calculate the absorptance (*A*), reflectance (*R*), and transmittance (*T*) using the scattering parameters (*S*-parameter) [60]. The definition of *S*-parameters, i.e., *S*_11_ and *S*_21_, are as follows: *S*_11_= (power reflected from the port 1/power incident on the port 1)^1/2^; and *S*_21_ = (power reflected from the port 2/power incident on the port 1)^1/2^. The dielectric constant of Ag is fitted by the Drude–Lorentz model [61,62], and the RI of the SiO_2_ layer is *n*_s_ = 1.45. We summarize the basic formulas used in this work in Table 1.

The proposed PMA can function as a temperature sensor. Ethanol (liquid state) is a good candidate because of its high RI temperature coefficient (i.e., d*n*/d*T* = 3.94 × 10^−4^, in the unit of *K*^−1^). The RI of ethanol can be expressed as [63]:*n* = 1.36048 − 3.94 × 10^−4^ (*T* − *T*_0_)(1)
where *T*_0_ = 20 °C and *T* is the ambient temperature. When ethanol is employed as the measuring liquid, the sensor's operating temperature range is −114.3 ≤ *T* ≤ 78 °C [64]. Table 1 shows the basic formulas used in this work.

The *S*-parameters of *R*(ω) = |*S*_11_(ω)|^2^, *T*(ω) = |*S*_21_ (ω)|^2^, and *A*(ω) = 1 − *R*(ω) − *T*(ω) denote the reflectance, transmittance, and absorptance, respectively. Δλ and Δ*n* are *λ_res_* shift and the difference in the RI. Additionally, *A*_max_ and *A*_min_ describe the maximum and minimum absorptance, respectively.

Thanks to the advanced technology of nanofabrication in recent years, the designed PMA sensor can be implemented [69]. The Ag-shell nanorods can be formed by preparing metal films with a periodic array of hollow nanorods via an inexpensive and versatile colloidal lithography technique through topologically continuous films [70]. However, this work focuses on the designed issue of PMA sensors with multiple channels in the visible range. As an alternative, several important works within the literature that involve the in-depth fabrication of PMA nanostructures are given [71,72,73].

## 3. Results and Discussion 

### 3.1. Comparison of the Optical Performance of Case 1 and Case 2

We adopt the geometrical parameters of the case 1 and case 2 PMAs in Table 2.

Figure 2a exhibits the absorptance spectra of case 1 (black line) and case 2 (red line) at the surrounding medium of *n* = 1.00 for x-polarization. We set the surrounding medium of *n* = 1.00 to compare the optical performance between case 1 and case 2. Table 3 and Table 4 also summarize the λ_res_ (nm), *FWHM* (nm), *A* (%), *Q* factor, and dipping strength (Δ*D*) of case 1 and case 2 at their corresponding modes 1–3 for x-polarization and y-polarization, respectively. As observed, the difference in absorptance peaks strongly relies on the plasmonic modes that occurred in the PMAs. There are three modes of absorbance peaks in cases 1 and 2, ranging in visible light. Compared to the case 1 PMA, the case 2 PMA reveals better optical performance because of a higher *A*, *Q* factor, and ∆*D*. The reason for this is that the air cavities in Ag-shell nanorods can raise the cavity plasmon resonance (CPR) effect [74,75], forming a higher SPP source than that of case 1. The SPPs of the case 2 PMA can help suppress the absorption losses in the designed PMA system because of a higher absorptance. The physical original can be attributed to the effective increase in the capacitance and inductance of the resonant PMA [76], which increases light–matter interactions in the plasmonic system.

As shown in Figure 2b and Table 4, the optical response of y-polarization is like those of x-polarization in mode 1 and mode 2. However, the absorptance spectrum curve of mode 3 for y-polarization reveals an oscillation pattern, leading to a smaller ∆*D* and a bigger FWHM. Therefore, we investigate the x-polarization cases in the subsequent simulations.

To understand the physical original, Figure 3a,b illustrates the x-polarized E-field intensity distributions at the *x*-*z* plane of modes 1–3 for the case 1 and case 2 PMAs. The solid Ag/Ag-shell nanorods could minimize the reflection by impedance matching, and the Ag layer at the bottom can mediate the resonant plasmonic mode conditions in the PMAs [77]. As seen in Figure 3a,b, the majority of the E-field exists among the solid Ag/Ag-shell nanorods and shows a substantial LSPR effect around the Ag surfaces. CPR mode happens in the Ag-shell nanorod cavities, and GPR mode occurs among the gaps of solid Ag/Ag-shell nanorods. They show a remarkable in-plane E-field enhancement in the gaps and holes, while a significant out-plane edge enhancement surrounds the PMA’s sides [78]. Note that the E-field distributions at the two sides have arisen from the gaps of the neighboring solid Ag/Ag-shell nanorods. These SPP modes could produce the constructive light–matter interactions in the PMA, leading to inductance and capacitance effects [76,79] in the plasmonic system. 

The enhanced E-field distribution can be described by the (+ −) charge pairs. Figure 4a,b shows the x-polarized surface charge density distribution and arrow surface of the current density (black arrows) of case 1 and case 2 at corresponding modes 1–3 for x-polarization, respectively. In case 1 and case 2, the (+ −) charge pairs only spread on the surface of solid Ag/Ag-shell nanorods, including the Ag layer at the bottom. Note that the case 2 PMA has more (+ −) charge pairs than case 1 because of the inner Ag-shell surface in the cavity of the PMA. Significantly, the plasmonic effects of the case 2 PMA can be governed by the hybridization of the plasmonic modes induced by SPR, GPR, and CPR, giving a more remarkable mutual inductance and dipolar effect on the PMA’s surfaces and capacitive coupling in the PMA’s cavity.

### 3.2. Structural Parameter Optimization of Case 2 PMA

Next, we examine the influence of geometrical parameters on the absorptance spectrum of the proposed case 2 PMA for x-polarization, including *h*, *t_Ag*, *r_1_*, *P_x_*, *θ*, and *φ.* Using Table 2, we vary one parameter to keep the other parameters constant. Figure 5a,b reveals the absorptance spectra with scanned parameters *h* and *t*_*Ag* for the case 2 PMA. The increasing *h* can expand the *z*-direction cavity capacity in the cavity of the case 2 PMA. Different cavity heights in the Ag-shell nanorods toward a vertical direction (varying *h*) lead to the change in plasmonic modes that could induce other resonance conditions in the case 2 PMA. In Figure 5a, the *h* parameter varies from 500–1300 nm with an interval of 25 nm. With the increase in *h*, the absorptance peaks show one to three modes in the full range of *h* values because of the different resonance conditions generated in the PMA system, leading to various aspects of plasmon modes on the metal surface. As Figure 5a clearly shows, three channels of absorptance modes can be obtained for *h* in the ranges of 995–1075 nm and 1215–1260 nm, while a broad range of two channels of absorptance modes are acquired for *h* in the ranges of 505–575 nm, 620–770 nm, and 915–995 nm. Based on Figure 5a, the *h* value can dominate the plasmonic effect in the proposed PMA. 

The absorptance spectrum for the case 2 PMA with varying *t*_Ag in the range of 10–80 nm, with increments of 5 nm, is investigated as shown in Figure 5b. As is shown in Figure 5b, three channels of absorptance modes can be obtained for *t*_Ag in a wide range of 28–65 nm. Here, the band linewidth of the coupled photonic–plasmonic resonance is associated with the Ag-shell nanorods with the cavity, which is related to the *Q*-factor, because of the modified photonic density of states and the modified radiative damping rate [80].

The other important parameters of the proposed PMA are *r_1_*, *P_x_*, *θ*, and *φ*. Different cavity sizes in the Ag-shell nanorods towards the transverse axis (i.e., changing *R* in the *x*-direction) are obtained by varying *r_1_*. The changing *r_1_* results in a change in cavity capacity in the case 2 PMA and a variation in surface charge distribution on the metal surface. The absorptance spectrum for the case 2 PMA with varying *r_1_* in the range of 70–200 nm, in increments of 5 nm, is investigated as shown in Figure 6a. Figure 6a shows three channels of absorptance modes that can be obtained for *r_1_* in the 88–120 nm range. Pitch is also an important parameter that can affect the absorptance of the proposed PMA. *P_y_* has less influence on the absorptance spectrum based on the FEM simulations. Therefore, we investigate the *P_x_* on the absorptance spectrum with *P_x_* values in the range of 600–1400 nm, with increments of 20 nm, as shown in Figure 6b. As seen, there are three channels of absorptance modes for *h* in the range of 850–1020 nm, while a broad range of two channels of absorptance modes are acquired for *h* in the range of 1020–1160 nm.

Figure 7a shows the absorptance spectrum for the case 2 PMA with varying *θ* in the range of 0–90°, with an interval of 5°. In Figure 7a, three and two channels of absorptance modes can be obtained for *θ* in the ranges of 0–11° and 25–85°, respectively. The other values of *θ* cannot effectively form the available resonance modes in the proposed plasmonic system. This can be intuitively explained by considering the multiple and asymmetric scattering originating from the oblique incidence. Additionally, the PMA structure demonstrates good tolerance of the operation angle of polarization (*φ*) at a wide range of incident angles (0–90°), as shown in Figure 7b.

### 3.3. Refractive Index and Temperature Sensor Performance of Case 2 PMA

The proposed case 2 PMA can serve as an RI sensor. Figure 8 depicts the absorptance spectrum of case 2 PMA under different surrounding media of *n* = 1.33, 1.35, and 1.37 for mode 1 to mode 5, respectively. Figure 9 shows the linear relationship curve between the ambient RI (*n*) and the λ_res_. All curves reveal linear relations with the resonance wavelength versus RI. As shown in Figure 8, the λ_res_ are red-shifted with the increasing RI. It is worth noting that the five absorptance modes have two additional modes compared to Figure 2a. When the RI values of *n* increase from the value of 1.00 in Figure 2a to 1.33 to 1.37 in Figure 8, the resonance wavelength red-shifts to a longer wavelength due to the increase in the RI of the surrounding medium. Therefore, the resonance modes in the shorter wavelength range (i.e., λ_res_ < 500 nm) shift to the investigated wavelength range of 500–850 nm, as shown in Figure 8. Based on Figure 9, the calculated sensitivity and *FOM* (*S*, *FOM*) for modes 1–5 can reach (600.00 nm/RIU, 120.00), (600.00 nm/RIU, 120.00 RIU^−1^), (600.00 nm/RIU, 120.00 RIU^−1^), (400.00 nm/RIU, 50.00 RIU^−1^), and (350.00 nm/RIU, 25.00 RIU^−1^), correspondingly. 

The distributions of EM power loss density and magnetic flux density in the unit–cell structure can offer detailed information concerning absorption in the proposed case 2 PMA system. Figure 10a–e shows the selected example (the case of *n* = 1.37) of the three-dimensional (3D) distribution of EM power loss density and magnetic flux density (in the unit of Tesla, streamlines, in cyan color) in the unit–cell structure at resonance modes for modes 1–5: as follows (a) λ_res_ = 812 nm (mode 1); (b) λ_res_ =764 nm (mode 2); (c) λ_res_ =708 nm (mode 3); (d) λ_res_ = 596 nm (mode 4); and (e) λ_res_ = 526 nm (mode 5). It is evident from Figure 10a–e that the EM power loss density is mainly distributed on the surface of the Ag-shell nanorods of the case 2 PMA structure at resonance, which is consistent with the distributions of the electric fields (Figure 3) and the surface charge density distribution (Figure 4). The distribution of EM power loss density is enormously dependent on the optical path on the surface of the metal-shell nanorods due to the combination of the guided mode with a different order in the air gaps and the surface plasmon polaritons mode on the Ag-shell nanorods and the thin Ag layer substrate [53]. 

As observed, for longer resonance wavelength at mode 1 (λ_res_ = 812 nm), the EM power loss density is mainly distributed on the side surface of Ag-shell nanorods, while the EM power loss density of mode 2 (λ_res_ = 764 nm) and mode 3 (λ_res_ = 708 nm) distributes on the top semi-spherical surface and splits into one and two segments on the side surface of the Ag-shell nanorods. For a shorter resonance wavelength at mode 4 (λ_res_ = 596 nm) and mode 5 (λ_res_ = 526 nm), the distributions of power loss density become more complex, i.e., the EM power loss density splits into more sections on the side surface of the Ag-shell nanorods due to the shorter optical path of mode 4 and mode 5. The magnetic flux density in the unit–cell structure at resonance modes for modes 1–5 show different degrees of binding on the surface of the Ag-shell nanorods and the gaps between Ag-shell nanorods, displaying that the high density of spiral streamlines encloses the Ag-shell nanorods along the *x*-direction and *y*-direction. This feature leads to the strong absorption of an incident EM wave and the enhancement of the coupling magnetic field, resulting in close interaction with the case 2 PMA, verifying that the surface current can induce on the metal surface. The results from Figure 10 suggest a new approach to obtain a multispectral response by integrating various plasmonic modes in a unit cell of the proposed case 2 PMA, which offers potential applications in nanophotonic devices, including optical sensors [81,82].

Except for the RI sensing, as investigated in Figure 8, the proposed case 2 PMA can also function as a temperature sensor. Using Equation (1), we can examine the absorptance spectrum versus temperature in the PMA structure. The structural parameters of Table 2 are used in the simulations. Figure 11a,b depicts the shift in the absorptance spectrum for temperatures ranging from −60 to 60 °C, in increments of 20 °C. The calculated RIs of ethanol from Equation (1) for different temperatures are as follows: *n* = 1.3920, 1.37642, 1.3762, 1.3684, 1.36048, 1.3526, and 1.34472 for temperature *T*= −60, −40, −20, 0, 20, 40, and 60 °C, correspondingly. The λ_res_ blue-shifts with the rising temperature because of the decrease in RI with the increasing temperature. Figure 11 shows how varying the temperature affects the λ_res_. The wavelength shifts are 26 nm, 26 nm, 26 nm, 20 nm, and 18 nm for modes 1–5 when the temperatures vary from −60 °C to +60 °C, in increments of 20 °C. The obtained temperature sensitivity *S_T_* = 0.22, 0.22, 0.22, 0.17, and 0.15 nm/°C ranges between 500–800 nm for modes 1–5, respectively. As seen in Figure 12, the proposed case 2 PMA also exhibits a linear relationship with an increase in the temperature range from −60 to +60 °C.

Many research works have used ethanol as a temperature-sensitive liquid in plasmonic materials for temperature sensors (e.g., [83,84,85,86,87]). The temperature resolution of the proposed PMA sensor can be defined as *R_T_* = ΔT Δ*λ_min_*/Δ*λ_peak_* [83,88]. For example, the variation of the temperature (Δ*T*) and Δ*λ_peak_* for mode 1 shown in Figure 11 is 20 °C and 26 nm, respectively. The wavelength resolution of the detector is assumed to be Δ*λ_min_* = 0.1 nm (see Refs. [83,89]). Consequently, the temperature resolution of the designed PMA sensor for mode 1 is 0.077 °C. 

To test the performance of the proposed PMA sensor as a RI or a temperature sensor, the detecting medium (or solution) can be infiltrated on the sensor’s surface. Based on [90], one can inspect the RI and the temperature sensing function independently or simultaneously by using the hybrid sensing probe to avoid the inherent interference of the RI. The robustness and stability of the proposed PMA is a vital factor that is beyond the scope of this article. The PMA can survive because only a very small amount of ethanol liquid is immersed [91], and the pressure influenced by the ethanol liquid can be ignored.

The utilization of plasmonic material inside the PMA could cause ohmic losses that raise the working temperature. The resulting heat source might heat the ethanol and change the environmental temperature. To avoid overheating the PMA, we should consider the maximum power of the incident light depending on the ethanol’s temperature coefficient (i.e., d*n*/d*T* = 3.94 × 10^−4^) in a real experiment. Because of a small bandgap in the semiconductor, Kriegner et al. [92] and Gandolfi et al. [93] utilized the InAs and InSb nanowires to mitigate the power of incident light that is absorbed in the PMAs. 

To show the excellent performance of the presented PMA sensor, Table 5 compares the sensor performance of the designed PMA structure with the similarly reported literature.

## 4. Conclusions

In conclusion, we proposed a multiple-channel PMA sensor based on a compact structure consisting of an array of hexagonal Ag-shell nanorods in a unit cell for RI and temperature-sensing applications. We explained the physical nature through the absorptance spectrum, E-field intensity, electromagnetic power loss density, magnetic flux density, and surface charge distribution based on FEM simulations. The absorptance spectrum of the designed PMA sensor can be tuned to have three or five absorptance modes. The recorded sensitivity (*S*) and *FOM* for modes 1–5 can obtain (600.00 nm/RIU, 120.00), (600.00 nm/RIU, 120.00 RIU^−1^), (600.00 nm/RIU, 120.00 RIU^−1^), (400.00 nm/RIU, 50.00 RIU^−1^) and (25.00 nm/RIU, 120.00 RIU^−1^), respectively. In addition, temperature sensitivity can simultaneously reach 0.22 nm/°C for modes 1–3. This work provides guidance for the design of the multiple-channel PMA and has excellent potential application as a RI or temperature sensor working in the visible range.

## Figures and Tables

**Figure 1 micromachines-14-00340-f001:**
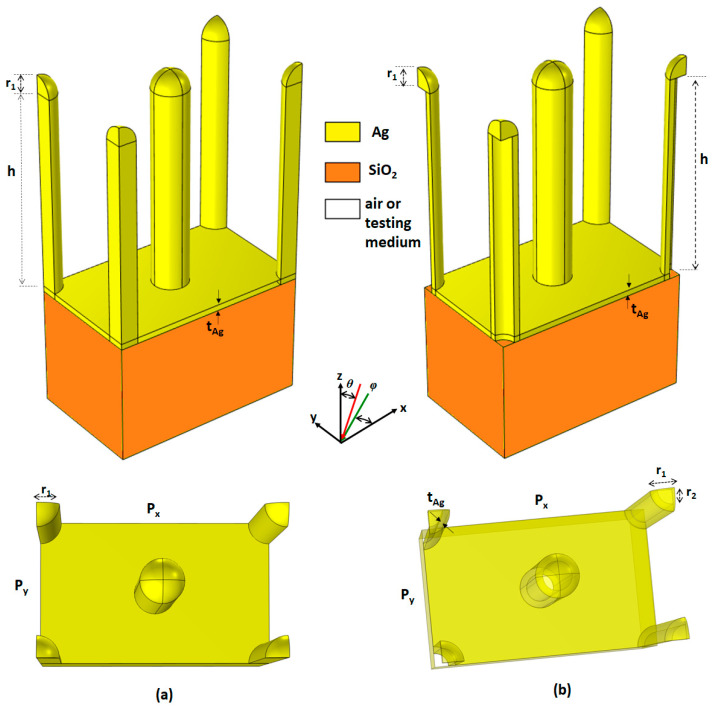
Unit cells and structural parameters of the designed PMAs: (**a**) case 1 PMA and (**b**) case 2 PMA. Top panels for side view and bottom panels for top view.

**Figure 2 micromachines-14-00340-f002:**
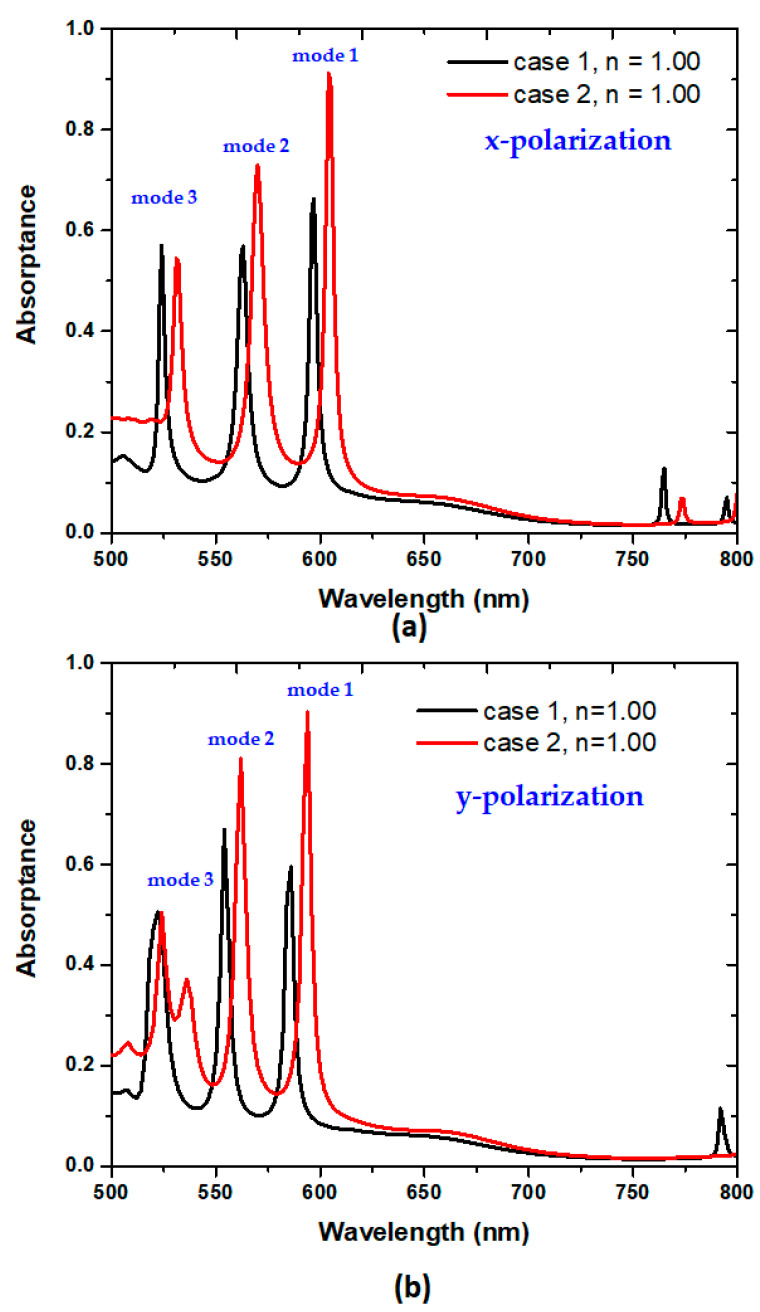
Absorptance spectra of the designed PMAs at surrounding medium *n* = 1.00 of case 1 (black line) and case 2 (red line) for (**a**) x-polarization and (**b**) y-polarization, respectively.

**Figure 3 micromachines-14-00340-f003:**
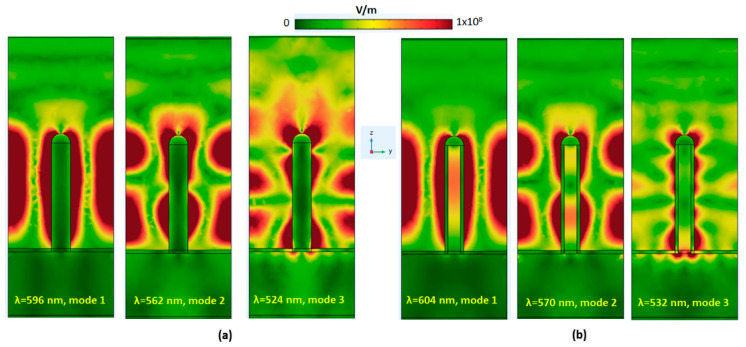
The x-polarized E-field intensity distributions at the *x*-*z* plane of modes 1–3 for (**a**) case 1 PMA and (**b**) case 2 PMA. The *x*-*z* plane intersects the central part of the central solid Ag/Ag-shell nanorods. The color scale corresponds to the norm of the electric field.

**Figure 4 micromachines-14-00340-f004:**
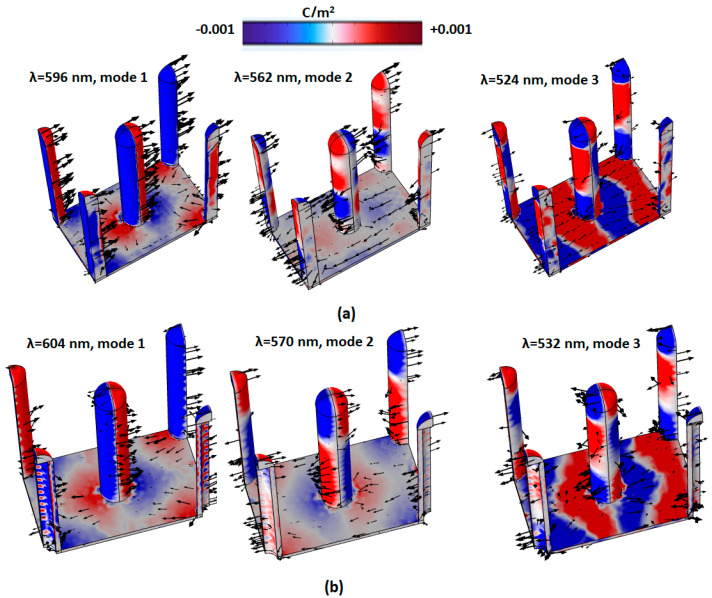
The x-polarized surface charge density distribution and arrow surface of current density (black arrows) of (**a**) cases 1 and (**b**) case 2 for modes 1–3, respectively.

**Figure 5 micromachines-14-00340-f005:**
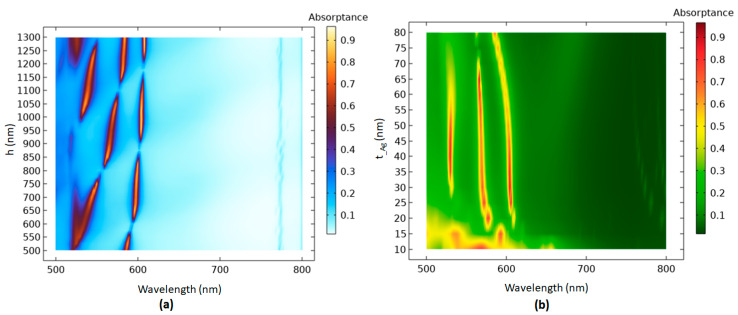
Absorptance spectra with scanned parameters (**a**) *h* and (**b**) *t*_*Ag* for the case 2 PMA.

**Figure 6 micromachines-14-00340-f006:**
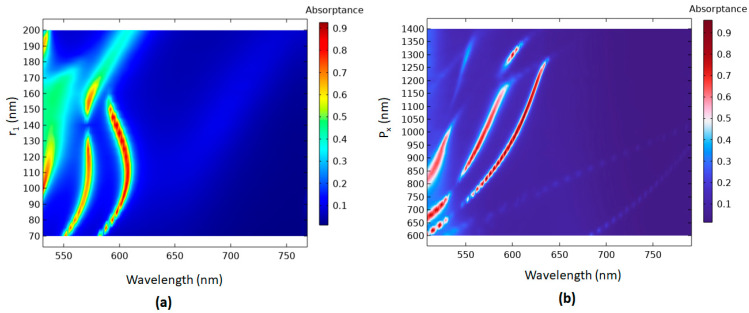
The absorption spectra with scanned parameters (**a**) *r_1_* and (**b**) *P_x_* for the case 2 PMA.

**Figure 7 micromachines-14-00340-f007:**
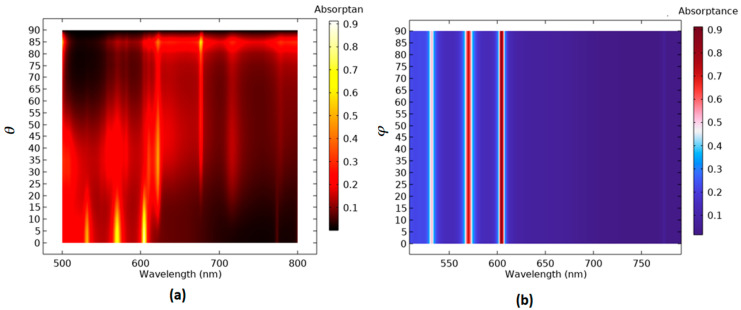
The absorption spectra with scanned parameters (**a**) *θ* and (**b**) *φ* for the case 2 PMA.

**Figure 8 micromachines-14-00340-f008:**
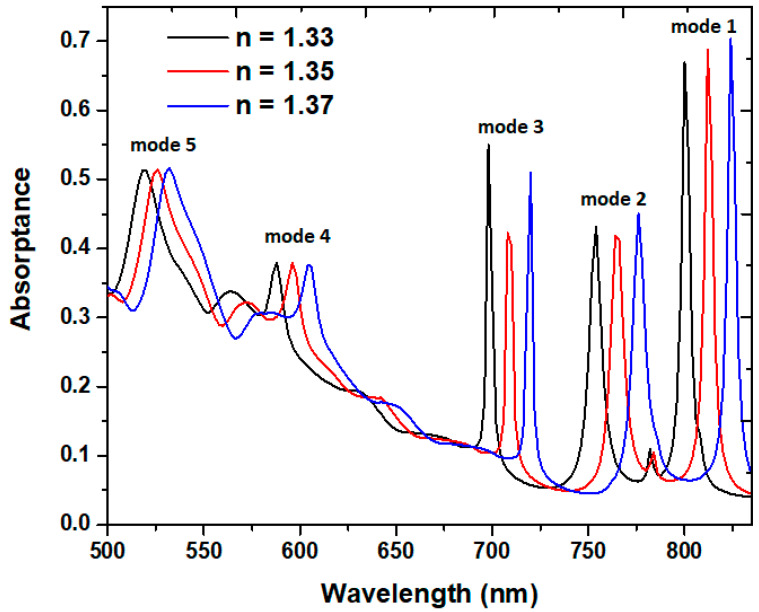
Absorptance spectrum of the case 2 PMA versus RI of *n* = 1.33, 1.35, and 1.37 for modes 1–5, respectively.

**Figure 9 micromachines-14-00340-f009:**
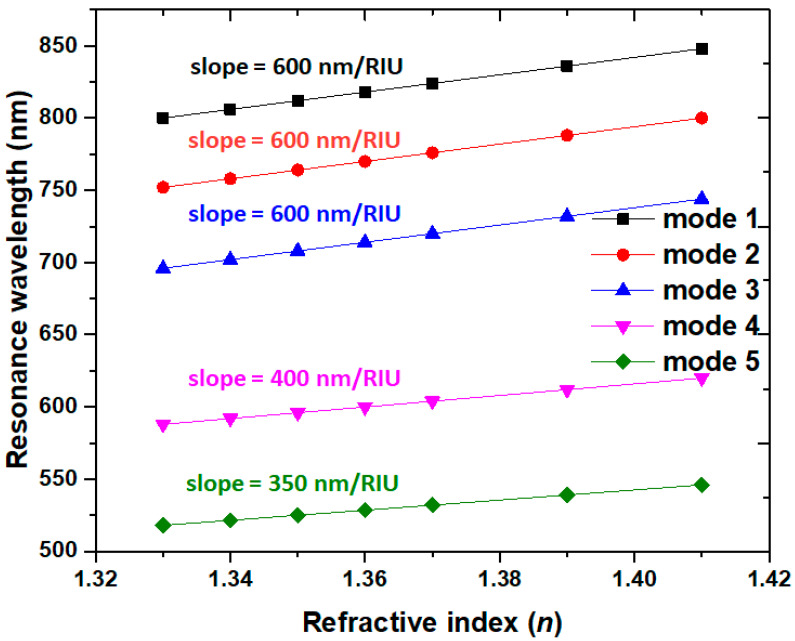
Resonance wavelength (λ_res_) versus the ambient RI (*n*) of case 2 PMA.

**Figure 10 micromachines-14-00340-f010:**
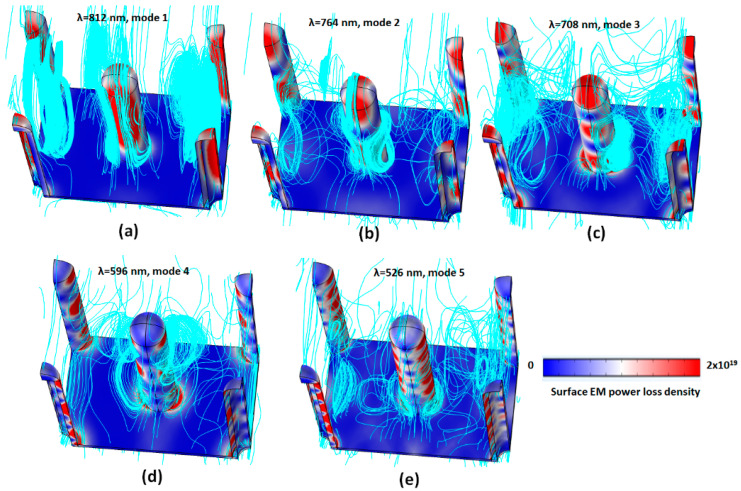
Selected example (the case of *n* = 1.37) of the three-dimensional (3D) distributions of EM power loss density and magnetic flux density (in the unit of Tesla, streamlines, in cyan color) in the unit–cell structure at resonance modes of case 2 PMA for modes 1–5: (**a**) λ_res_ = 812 nm (mode 1) (**b**) λ_res_ = 764 nm (mode 2) (**c**) λ_res_ = 708 nm (mode 3) (**d**) λ_res_ = 596 nm (mode 4) and (**e**) λ_res_ = 526 nm (mode 5), respectively.

**Figure 11 micromachines-14-00340-f011:**
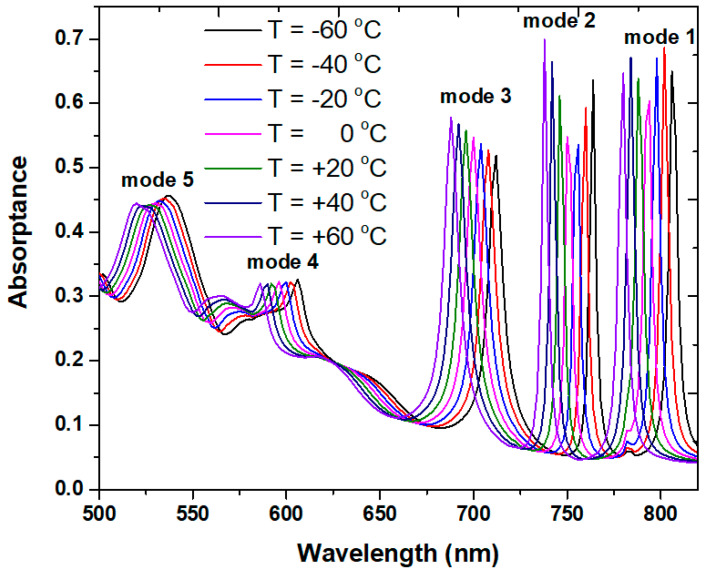
Absorptance spectrum of case 2 PMA for temperatures ranging from −60 to 60 °C in increments of 20 °C.

**Figure 12 micromachines-14-00340-f012:**
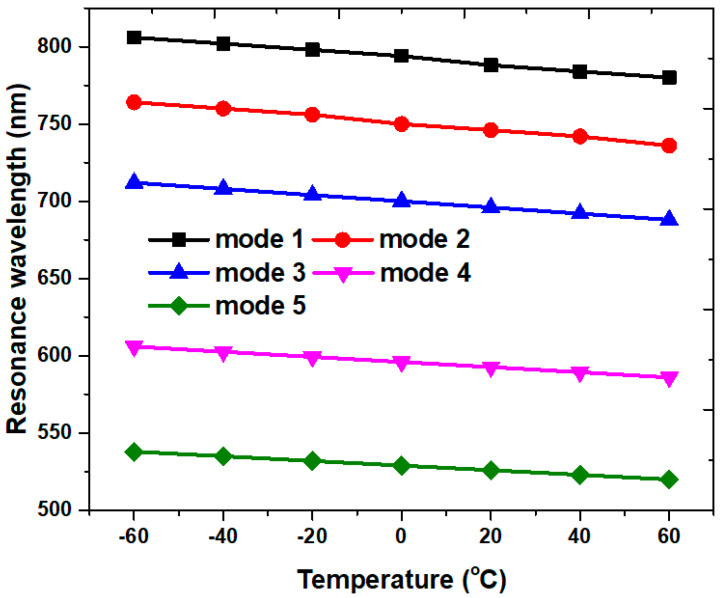
Resonance wavelength versus temperatures of case 2 PMA ranging from −60 to 60 °C in increments of 20 °C.

**Table 1 micromachines-14-00340-t001:** Basic formulas used in this work.

Name	Formula	Unit
*A* (absorptance) [11]	A(ω) = 1 − *R*(ω) − *T*(ω)	
*S* (sensitivity) [65]	*S* = Δλ/Δ*n*	nm/RIU
*FOM* (figure of merit) [66]	*FOM* = *S*/*FWHM*	1/RIU
*Q-factor* (quality factor) [67]	*Q* = λ_res_/FWHM	
Δ*D* (dipping strength) [68]	Δ*D* = (*A*_max_ − *A*_min_) × 100%	
*S_T_* (temperature sensitivity)	*S_T_* = Δλ/Δ*T*	nm/°C

**Table 2 micromachines-14-00340-t002:** Structural parameters of the proposed PMA.

*h* (nm)	*P_x_* (nm)	*P_y_* (nm)	*t_Ag* (nm)	*r_1_* (nm)	*r_2_* (nm)	*θ*(°)	*φ*(°)
1000	1000	P*_x_*/(3)^1/2^	30	90	*r_1_* − *t_Ag*	0	0

**Table 3 micromachines-14-00340-t003:** The *λ_re_*_s_, *FWHM*, *A* (%), *∆D*, *Q* factor, Sensitivity (*S*), and *FOM* of the case 1 and case 2 PMAs at their corresponding resonance modes for x-polarization.

	Case 1				Case 2	
mode	1	2	3	1	2	3
λ_res_ (nm)	596	562	524	604	570	532
FWHM (nm)	5	4	4	5	4	3
A (%)	64.10	54.7	57.05	91.24	73.03	53.61
∆D (%)	56.90	45.40	47.20	81.22	59.20	39.55
*Q* factor	120.40	140.50	131.00	120.80	142.50	177.33

**Table 4 micromachines-14-00340-t004:** The *λ_re_*_s_, *FWHM*, *A* (%), *∆D*, *Q* factor, Sensitivity (*S*), and *FOM* of the case 1 and case 2 PMAs at their corresponding resonance modes for y-polarization.

	Case 1				Case 2	
mode	1	2	3	1	2	3
λ_res_ (nm)	586	554	522	594	562	536
FWHM (nm)	5	4	8	5	4	8
A (%)	50.06	54.7	50.06	90.334	81.092	37.233
∆D (%)	48.41	65.94	49.05	89.296	80.93	35.66
*Q* factor	117.20	138.50	65.25	118.80	140.50	67.00

**Table 5 micromachines-14-00340-t005:** Comparison of the sensor performance of the proposed PMA with similar works.

Reference/Working Wavelength (nm)	Resonator Structure	Mode Number	RI Sensitivity *S* (nm/RIU)	Max. FOM (RIU^−1^)	Temperature Sensitivity *S_T_* (nm/°C)
[33]/740–785	Au/dielectric/Au layers	1	425	233.5	NA
[35]/900–1500	Ag/dielectric/Ag layers	2	630, 465	NA	NA
[34]/400–800	dielectric nanodisks	2	NA	NA	NA
[53]/600–1500	vertical-square-split-ring	3	1194,816,473	16.17	NA
[94]/900–1350	dielectric–dielectric–metal	3	474,463,255, 770	338.6	NA
This work	hexagonal Ag-shell nanorods	5	600, 600, 600, 400, 350	120.0	0.22,0.22,0.22 0.17, 0.15

## Data Availability

Not applicable.

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
