# Peer review of "Visible-Range Multiple-Channel Metal-Shell Rod-Shaped Narrowband Plasmonic Metamaterial Absorber for Refractive Index and Temperature Sensing"

_micromachines, 2023, doi:10.3390/mi14020340_

Round 1
Reviewer 1 Report
This paper is a good and instructive piece of work. Overall, I recommend this paper to be accepted with following minor revisions.
1. It would be good for authors to briefly discuss how these resonance response of perfect absorbance is utilized in RI and temperature sensing applications.
2. I think authors should put more details on how the simulations are performed. For example, how they excite the energy, simulate A, R and T etc.
3. Detailed background physical mechanisms must be provide for each simulation results, rather than only simple description of simulation results itself.
4. Why the thickness of the samples to be detected is set as 30nm?
5. What the authors have studied are all based on x-polarized incident waves, how about the y-polarized waves?
6. Comparisons between the proposed design and the recently reported literatures should be given in order to demonstrate the innovation and the breakthrough of the current design and its performances.
Reviewer 2 Report
Please find my comments in the attached .pdf file

Reviewer 3 Report
Comments on the manuscript
In this manuscript entitled “Visible-range multiple channels metal-shell rod-shaped narrow band plasmonic metamaterial absorber for refractive index and temperature sensing,” the authors presented a systematic study on plasmonic metasurfaces supporting multiple resonances, which is driven by the physics of FP interference. They also included index/temperature sensing simulation results. Overall, the manuscript, in general, is interesting and their simulation results are of interest to the field of plasmonic metasurfaces. Also, the manuscript is technically sound with well-supported conclusions and assertions.
Thus, this manuscript meets the scope of micromachines I recommend publishing this work after the following comments are addressed.
1. My major concern is the motivation of the manuscript. The unit-cell structure design is new. However, it is of an ultra-high aspect ratio, which is often very difficult in real experiments. Thus, the fabrication difficulty reduces its practical implementation possibility. Have the authors thought about the potential fabrication scheme despite that it is a simulation-only work? Please comment. Also, how can the multiple resonance metasurface sensors outperform their single particle sensors in the visible wavelength [Celiksoy, Sirin, et al. "Intensity-Based Single Particle Plasmon Sensing." Nano Letters 21.5 (2021): 2053-2058]? Please discuss.
2. In the abstract. “The novelty of the proposed PMA is that it can tune either three or five absorptance channels with a narrow band in the visible range. The recorded sensitivity and figure of merit (S , FOM) for modes 1 – 5 can obtain (600.00 nm/RIU, 120.00), (600.00 nm/RIU, 120.00 RIU-1), (600.00 nm/RIU, 120.00 RIU-1), (400.00 nm/RIU, 120.00 RIU-1) and (350.00 nm/RIU, 120.00 RIU-1), respectively.” Firstly, avoid using “novelty” unless it is something completely new or abnormal. Secondly, claiming sensitivity and figure of merit without any comparations is meaningless. Please be careful.
3. “After then, a variety of works have been proposed to concern the absorptance response of PMA working at the frequency range from microwave [29] through terahertz(THz)[30], infrared[31], and into the visible region[32-34].” Some perfect absorbers at mid-infrared regimes are not mentioned at all. For example (1) full-stokes polarization perfect absorption; (2) dark mode perfect absorbers [Liang, Yao, et al. "Bound states in the continuum in anisotropic plasmonic metasurfaces." Nano Letters 20.9 (2020): 6351-6356]. Better to include them.
4. Figure 1. It seems that the two images at the bottom are screenshots of COMSOL. This is not professional. Better to redraw them if the authors can use other 3D drawing software.
5. Have the authors considered the angular stability of different resonance modes at various oblique incidences?
Round 2
Reviewer 2 Report
Please find my comments in the attached .pdf file.
Thank you very much for your attention.
Best Regards.

Round 3
Reviewer 2 Report
The paper has improved and now it deserves publication as it is.